# Systemic signaling contributes to the unfolded protein response of the plant endoplasmic reticulum

Ya-Shiuan Lai[1,2], Giovanni Stefano[1,3], Starla Zemelis-Durfee[1,3,4], Cristina Ruberti[1], Lizzie Gibbons[1] & Federica Brandizzi[1,3,4]

The unfolded protein response (UPR) of the endoplasmic reticulum constitutes a conserved and essential cytoprotective pathway designed to survive biotic and abiotic stresses that alter the proteostasis of the endoplasmic reticulum. The UPR is typically considered cell-autonomous and it is yet unclear whether it can also act systemically through non-cell autonomous signaling. We have addressed this question using a genetic approach coupled with micro-grafting and a suite of molecular reporters in the model plant species *Arabidopsis thaliana*. We show that the UPR has a non-cell autonomous component, and we demonstrate that this is partially mediated by the intercellular movement of the UPR transcription factor bZIP60 facilitating systemic UPR signaling. Therefore, in multicellular eukaryotes such as plants, non-cell autonomous UPR signaling relies on the systemic movement of at least a UPR transcriptional modulator.

[1] MSU-DOE Plant Research Lab, Michigan State University, East Lansing, MI 48824, USA. [2] Cell and Molecular Biology Program, Michigan State University, East Lansing, MI 48824, USA. [3] Plant Biology Department, Michigan State University, East Lansing, MI 48824, USA. [4] Great Lakes Bioenergy Research Center, Michigan State University, East Lansing, MI 48824, USA. Correspondence and requests for materials should be addressed to F.B. (email: fb@msu.edu)

In physiological conditions of growth and in disease, eukaryotic life depends on the biosynthetic ability of the endoplasmic reticulum (ER) to synthesize correctly folded secretory proteins. Conditions that alter the ER proteostasis and induce accrual of misfolded proteins in the ER lead to a potentially lethal condition known as ER stress[1,2]. At the onset of ER stress, cells activate cell-intrinsic UPR signaling pathways mediated by specialized ER stress sensors whose function is to reprogram gene expression for the synthesis of effectors that attenuate ER stress and restore the biosynthetic ability of the ER[3]. If the UPR is ineffective to initiate proper cytoprotective mechanisms to attenuate ER stress, it induces programmed cell death[4].

The main branch of the UPR is mediated by the ER-associated kinase and ribonuclease inositol-requiring protein 1 (IRE1) through largely conserved mechanisms. During ER stress, upon oligomerization and trans-autophosphorylation for self-activation, IRE1 splices the mRNA of a basic leucine zipper (bZIP) transcription factor, namely HAC1 in yeast, XBP1 in metazoans, and bZIP60 in plants. This step removes the coding region for a transmembrane domain (TMD), releasing the translational inhibition of a potent UPR transcriptional factor. The newly synthesized transcription factor is translocated to the nucleus where it modulates the expression of nuclear UPR target genes for the restoration of ER proteostasis[5–8]. Multicellular eukaryotes also harness another UPR branch, which is mediated by ER membrane tethered transcription factors (MTTFs), such as ATF6 in metazoans and bZIP28 in plants[9]. Upon ER stress sensing, these MTTFs translocate to the Golgi where the transcription factor domain is cleaved off the transmembrane anchor and is then transported to the nucleus to regulate transcription of UPR target genes[9,10].

In addition to cell-intrinsic signaling, the metazoan UPR may actuate non-cell autonomous signaling for the activation of stress responses in tissues and cell types that are different from those where the ER stress signal is originated. Specifically, in *C. elegans* intercellular signaling of the UPR has been induced through the overexpression of spliced (i.e., active) XBP1 in neuron cells, which elicits UPR activation in non-stressed intestine cells[11]. Similarly, in mice overexpression of active XBP1 in hypothalamic proopiomelanocortin (POMC) neurons is followed by non-cell autonomous splicing of XBP1 and UPR activation in the liver[12]. Although the existence of secreted stress signals to actuate transcellular UPR has been hypothesized[11], the identity of the effectors that act downstream XBP1 in intercellular communication of the UPR in metazoans is currently unknown. It is yet also unknown whether the systemic UPR signaling occurs in experimental conditions that do not rely on tissue-specific overexpression of XBP1.

Plants show cell-intrinsic UPR signaling;[13] however, whether they also execute non-cell autonomous UPR signaling is still an open question. Here, we demonstrate that in plants, in addition to cell-autonomous signaling, the UPR extends to systemic tissues by non-cell autonomous signaling through the contribution of the mobile UPR transcription factor bZIP60. Our findings indicate that in eukaryotes non-cell autonomous UPR signaling can directly rely on the translocation of at least one UPR transcriptional regulator.

## Results

**Spliced bZIP60 translocates transcellularly.** To test whether systemic UPR signaling may take place in plants, we first adopted a cell-type specific expression assay in *Arabidopsis* transgenic roots. We used the short-root (SHR) promoter, which is exclusively active in the stele, the central tissue of the root, and drives the expression of SHR[14]. The latter is a nucleus-localized

transcription factor that moves from the stele, where it is synthesized, to the endodermis, a tissue layer surrounding the stele; notoriously, SHR does not reach the cortex and epidermis, which envelope the endodermis[14]. We used the *SHR* promoter to drive expression of cytosolic green fluorescent protein (GFP) (pSHR-GFP)[15,16], and GFP fused either to SHR (pSHR-SHR-GFP)[14] or to a constitutively active form of bZIP60, spliced bZIP60-GFP (pSHR-sbZIP60-GFP). We used wild-type Col-0 (hereafter Col-0), an *SHR* knockout[17] (*shr-2*, hereafter *shr*), and *bzip28/60-1*[18] (hereafter *bzip28/60*). In *bzip28/60* both UPR branches (i.e., bZIP60 and bZIP28) are inactive in conditions inducing ER stress (i.e., Tunicamycin (Tm) treatment), and the expression of UPR genes, including *BiP3*, the major target of sbZIP60[8], is not actuated[19]. This setup was therefore designed to test movement of bZIP60 across tissues. A GFP fusion of bZIP60 driven by the native promoter in a *bzip60* mutant[20] is localized throughout the root tissues in control conditions and in conditions of ER stress (Supplementary Fig. 1) hampering the possibility to assess systemic movement of this transcription factor. In our experimental setup, we expected that cytosolic GFP would be detected exclusively in the stele, while SHR-GFP would be localized in the stele and the endodermis. Conversely, if sbZIP60 moved transcellularly, then *sbZIP60-GFP* expression in the stele would result in the accumulation of sbZIP60-GFP in the stele as well as in other cell layers. Confocal imaging of cytosolic GFP and SHR-GFP in the root of the respective Col-0 and *shr* transgenic lines showed a diffuse distribution of cytosolic GFP in the stele, and a localization of SHR-GFP in the nuclei of the stele and endodermis (Fig. 1a). These results are consistent with earlier findings[21] and indicate that stele-expressed cytosolic GFP accumulates only in the stele, while SHR-GFP, which is produced in the stele, moves to the endodermis[15,22]. When we analyzed *bzip28/60; pSHR-sbZIP60-GFP* roots, we found accumulation of sbZIP60-GFP in the nuclei and cytoplasm of cells in the stele and endodermis, as well as cortex and epidermis (Fig. 1a), which is comparable with the localization of GFP-bZIP60 driven by the *bZIP60* native promoter in conditions of ER stress[20] (see also Supplementary Fig. 1). In addition, such distribution pattern was visible throughout the division, elongation and differentiation zones of roots with graded level of fluorescence from the younger regions of the root upward (Supplementary Fig. 2). In light of the restricted accumulation of cytosolic GFP to the stele and of SHR-GFP to the stele and endodermis, these results strongly support that sbZIP60 can move transcellularly from the stele to the epidermis through the endodermis and cortex.

Next, we tested whether the transcellular movement of sbZIP60 could play a role in UPR signaling in the tissues in which it is translocated. We developed a genetically encoded reporter for UPR activation by *sbZIP60* in systemic tissues by expressing β-glucuronidase (GUS) under the control of the *BiP3* promoter (pBiP3-GUS) in either Col-0 (*Col-0; pBiP3-GUS*; positive control) or *bzip28/60*. In the latter background we introduced *pBiP3-GUS* alone (*bzip28/60; pBiP3-GUS*; negative control) or in combination with *pSHR-sbZIP60-GFP* (*bzip28/60; pSHR-sbZIP60-GFP/pBiP3-GUS*). We then tested where the UPR could be activated systemically in the root by stele-expressed sbZIP60. We expected that if sbZIP60 activated the *BiP3* promoter in a systemic manner, then the tissue labeling by GUS would mirror the verified tissue distribution of sbZIP60-GFP (Fig. 1a) and would show staining intensity levels above background. As expected, we found no expression of *pBiP3-GUS* throughout the *bzip28/60* roots (Fig. 1b, c), owing to the lack of functional bZIP60 and bZIP28. In wild type, *BiP3* is generally expressed in the absence of induced ER stress[9,23]. Consistently with this, in *Col-0; pBiP3-GUS* we verified GUS expression, which was visible in the endodermis of the upper maturation zone, and predominantly in the stele and

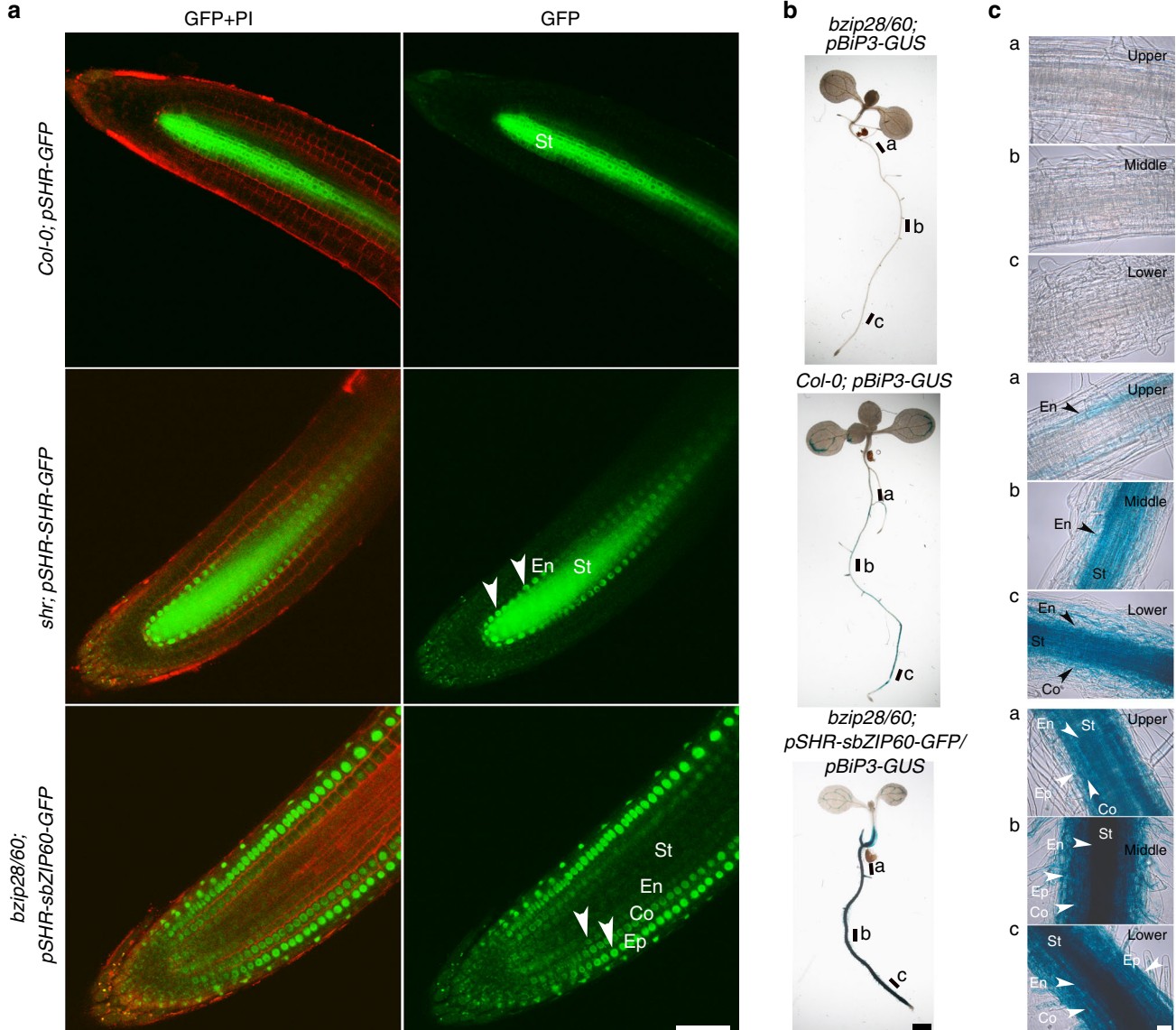

**Fig. 1** Intercellular translocation of sbZIP60 induces *BiP3* expression in systemic tissues. **a** Confocal laser scanning microscopy of *Col-0; pSHR-GFP*, *shr; pSHR-SHR-GFP*, and *bzip28/60; pSHR-sbZIP60-GFP* at the primary root tips of 5-day-old transgenics reveals stele (St) accumulation of GFP, and stele and endodermis (En) distribution of SHR-GFP; noticeably, sbZIP60-GFP is localized in the stele, endodermis, cortex (Co) and epidermis (Ep). Similarly to SHR-GFP, sbZIP60-GFP is localized in nuclei (arrows). As also reported earlier[22], we did not find SHR-GFP localization in the nuclei of the cortex and epidermis. Propidium iodide (PI) was used for counterstaining. Scale bar: 50 µm. **b** Expression of *pBIP3:GUS* in *bzip28/60*, Col-0 and *bzip28/60;pSHR-sbZIP60-GFP* seedlings grown vertically on half LS agar medium for 11 days. X-Gluc was used for histochemical staining to monitor GUS activity. Scale bar: 100 µm. **c** Longitudinal confocal optical sections of the regions along the primary root shown in **b**. Epidermis: Ep; Co: cortex; En: endodermis; St: stele. The indications upper, middle and lower refer to the a, b, and c zones indicated in panel **b**. Scale bar: 20 µm

endodermis of the middle maturation and lower maturation zones (Fig. 1b, c), in agreement with tissue-specific transcriptomics analyses of the root[24]. In conditions of ER stress, the *BiP3* expression was robustly enhanced in the stele, endodermis, cortex, and epidermis layers throughout all the root zones (Supplementary Fig. 3), consistently with previous findings[25]. We then analyzed *bzip28/60; pSHR-sbZIP60-GFP/ pBiP3-GUS* transgenic plants and found a strong GUS expression in the stele and endodermis but also in the cortex and epidermis of all the root zones under analysis (Fig. 1b, c). The strong GUS activity in the *bzip28/60; pSHR-sbZIP60-GFP/pBiP3-GUS* line is likely linked to the overabundance of sbZIP60 driven by pSHR in these tissues compared to *Col-0; pBiP3-GUS*. Importantly also, the GUS activity in *bzip28/60; pSHR-sbZIP60-GFP/pBiP3-GUS* mirrors the verified distribution of pSHR-sbZIP60-GFP in the cortex and

epidermis as well as *BiP3* expression pattern under stress condition (Fig. 1a, Supplementary Fig. 3). These results support our original observations that sbZIP60-GFP can translocate systemically from a tissue where it is specifically expressed and activate transcription of a target gene in systemic tissues. Taken together, these data indicate that, in conditions of overexpression of a constitutively active UPR modulator, the UPR signaling is executed systemically in plants. The results also indicate that overexpressed sbZIP60 can act as transcellular mobile transcription factor, triggering UPR gene expression in a systemic manner.

**Systemic induction of UPR genes**. To test a systemic signaling of sbZIP60 from the root to a different organ, we assessed the expression of UPR marker genes in the shoot and root of *bzip28/*

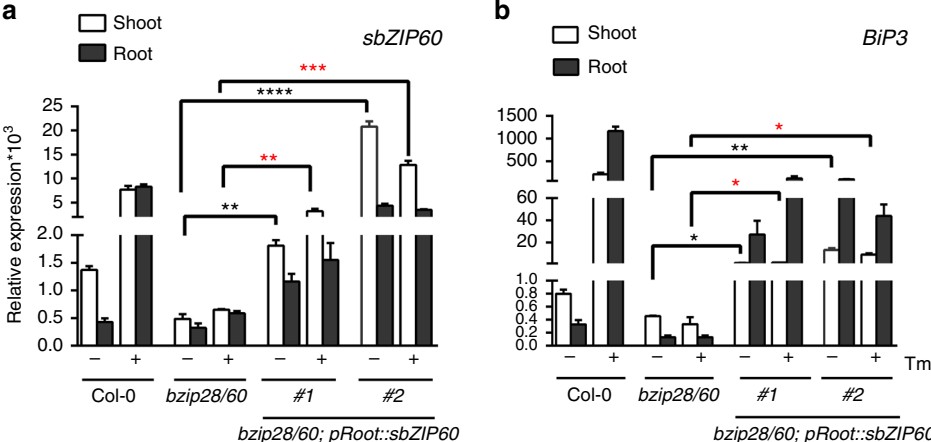

**Fig. 2** Root-expressed *sbZIP60* transcripts are translocated to the shoot. **a**, **b** Quantitative RT-PCR analyses of *sbZIP60* (**a**) and *BiP3* (**b**) in 14-day-old wild type (Col-0), *bzip28/60*, and *bzip28/60; pRoot::sbZIP60* seedlings treated with DMSO or 0.5 μM Tm (Tunicamycin) at the root in the shoot–root split system for 24 h. Transcription of *UBQ10* was used as internal control. Error bars represent s.e.m among three biological replicates. Data significantly different from the corresponding control are indicated by asterisks (*$P < 0.05$, **$P < 0.01$, ***$P < 0.001$, ****$P < 0.0001$; Unpaired *t*-test). Black asterisks designate differences between *bzip28/60* and *bzip28/60; pRoot::sbZIP60* lines under Tm non-treatment condition. Red asterisks designate differences between *bzip28/60* and *bzip28/60; pRoot::sbZIP60* under Tm treatment condition

*60* lines that express *sbZIP60* exclusively in the root. We expected that if a UPR signal moved shoot-ward from the root, we would detect expression of *sbZIP60*, and possibly of *BiP3*, not only in the roots but also in the shoots. To express sbZIP60 specifically in the root, we generated transgenic *bzip28/60* expressing *sbZIP60* under the control of the *pRoot* promoter[26]. *pRoot* drives the *Arabidopsis* glycosidase-transferase *GLYT* expression specifically in the roots[26], both in unchallenged and Tm-challenged seedlings, as supported by the evidence that the expression of *GLYT* in the shoots is proximal to the detection limit of quantitative RT-PCR (qRT-PCR) and not statistically different in both experimental conditions (Supplementary Fig. 4). We selected two independent transgenic lines that we named *bzip28/60; pRoot::sbZIP60* and adopted an *Arabidopsis* shoot–root split culture system[27] in which 2-week old intact seedlings grown on vertical plates are transferred to Petri dishes that are subdivided by a sealed plate divider to separate growth media with different composition[28]. By laying intact seedlings across the plate divider, this system exposes shoot and root to the medium contained in each plate sub-compartment separately. We applied Tm or DMSO (Tm solvent) for 24 h to the medium exposed to the root and compared the levels of UPR gene transcripts in shoot and root of seedlings challenged by Tm at the root by qRT-PCR. We used Col-0 and *bzip28/60* as positive and negative controls, respectively. As expected, in Col-0 the levels of the UPR gene transcripts *sbZIP60* and *BiP3* were more abundant in Tm-treated seedlings compared to DMSO control (Fig. 2a, b). Conversely, the *bzip28/60* mutant did not show elevation of either UPR marker gene in the absence and in the presence of Tm (Fig. 2a, b), despite its ability to absorb Tm (Supplementary Fig. 5). However, the *bzip28/60; pRoot::sbZIP60* lines showed *sbZIP60* and *BiP3* transcripts in the root at levels that were significantly higher compared to *bzip28/60* (Fig. 2a, b), indicating that the *pRoot* promoter is functional for the expression of these genes. Next, analyses of *sbZIP60* and *BiP3* transcripts in the shoot of *bzip28/60; pRoot::sbZIP60* lines showed significantly higher levels of *sbZIP60* and *BiP3* transcripts compared to *bzip28/60*. Unlike in Col-0, in *bzip28/60; pRoot::sbZIP60* the *sbZIP60* and *BiP3* transcript levels were largely unchanged by Tm treatment, as it would be expected for genes normally controlled by ER stress-responsive promoters. Together with the evidence that *pRoot* is unresponsive to Tm and that it is expressed specifically in roots (Supplementary Fig. 4), as

well as the consideration that the experiments were conducted in a genetic background normally lacking endogenous expression of the UPR bZIP-transcription factors and, consequently of their target genes under ER stress, these data indicate that the root-generated *bZIP60* and *BiP3* transcripts are found in tissues in which their expression is not driven locally by an ER stress-responsive promoter. These findings support the hypothesis that the plant UPR signaling is non-cell autonomous and imply that the transcellular translocation of at least *sbZIP60* transcripts from root to shoot may be involved in long-distance transduction of the UPR.

**Systemic UPR acts systemically in a shoot-ward direction**. We next tested the occurrence of endogenous systemic UPR using the shoot–root split culture system in order to apply Tm for 24 h either to the medium at the shoot or the root of 2-week old wild-type Col-0 seedlings (Fig. 3a); we then monitored the UPR signaling in each portion of the seedling by qRT-PCR (Fig. 3b). We first monitored the *sbZIP60* and *BiP3* mRNA levels in seedlings treated with Tm at the roots and found increased levels compared to the root of a mock control (Fig. 3b; D/T vs 0 h). When we analyzed the untreated shoot of the root-treated seedlings, we observed a significant raise in the transcript levels of *sbZIP60* and *BiP3* compared to mock control (Fig. 3b; D/T vs 0 h). These results indicate that Tm-treatment of the root leads to UPR signaling activation both in the Tm-treated root as well as in the untreated shoot. Next, we analyzed the UPR gene transcripts in seedlings treated with Tm at the shoots (Fig. 3a; T/D). We found that both *sbZIP60* and *BiP3* transcript levels increased in the shoot compared to mock control (Fig. 3b; T/D). Furthermore, in net contrast to the D/T seedlings, no significant induction of *sbZIP60* and only a slightly induction of *BiP3* were detected in the untreated root upon shoot treatment compared to mock control (Fig. 3b; T/D vs 0 h). These results point towards the possibility of a systemic actuation of ER stress responses mainly in a shoot-ward direction. We next tested the kinetic profiles of *bZIP60* transcription and splicing as well as the subsequent activation of *BiP3* expression in response to ER stress within 24 h. We found that in the Tm-treated roots, a significant induction of unspliced *bZIP60* (*unbZIP60*) occurred at 3 h (Supplementary Fig. 6A). Consistently, the emergence of *sbZIP60* transcripts started at 3 h and reached peak expression at 6 h post treatment followed by a

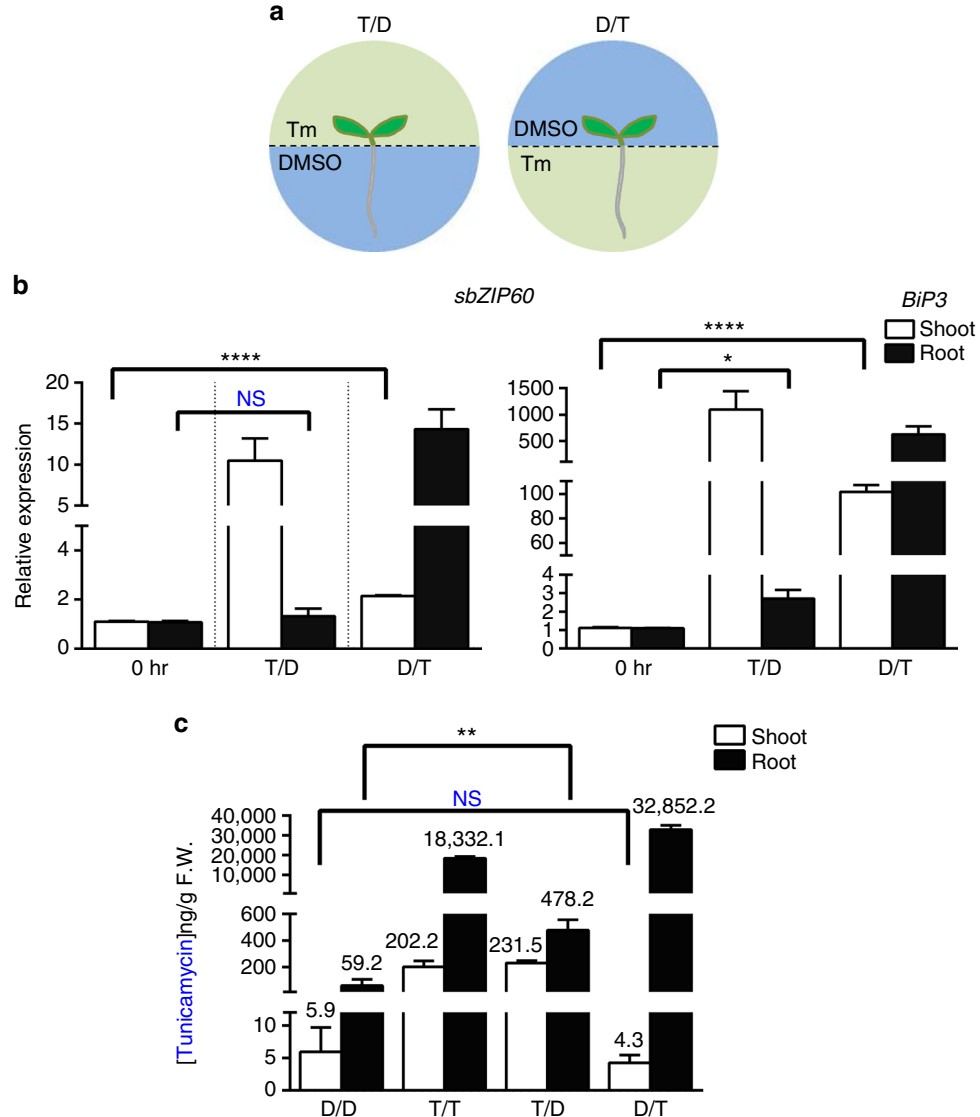

**Fig. 3** Local ER stress ignites the UPR systemically, mostly in a shoot-ward direction. **a** Diagrams illustrating the *Arabidopsis* shoot–root split culture system in which the shoot and root of an intact seedling are exposed to separate growth media with different chemical conditions: mock DMSO (D) or 0.5 μM Tm (T, Tunicamycin). T/D denotes shoot on Tm-containing medium and root on DMSO-containing medium; conversely, D/T denotes shoot on DMSO-containing medium and root on Tm-containing medium. **b** Quantitative RT-PCR analyses of UPR markers in 14-day-old wild-type seedlings treated with DMSO or 0.5 μM Tm for 24 h as described in **a**. Values are presented relative to non-treated control (0 h), which was set to 1. Transcription of *UBQ10* was used as internal control. Error bars represent s.e.m among three biological replicates. Data significantly different from the corresponding control are indicated by asterisks (*$P < 0.05$, ****$P < 0.0001$, NS, nonsignificant; Unpaired *t*-test). **c** Quantitative HPLC/MS analyses of Tm content in shoot and root of seedlings after treatments as described in **a** in a shoot–root split culture system. The numbers over the histograms express ng g$^{-1}$ fresh weight (F.W.). Data significantly different from the corresponding control are indicated by asterisks (**$P < 0.01$, NS, nonsignificant; Unpaired *t*-test)

decline of induction levels at 12 h and 24 h post treatment (Supplementary Fig. 6B). The *BiP3* expression displayed a similar trend like *sbZIP60* but reached the peak at 12 h post treatment (Supplementary Fig. 6C), in line with a time correlation of the *sbZIP60*-driven *BiP3* activation in response to the ER stress. In the untreated shoots, a significant induction of *unbZIP60* and *sbZIP60* was detected at 3 h post treatment and *BiP3* expression was predominantly detected at 6 h post treatment (Supplementary Fig. 6A–C) supporting a roughly 3 h-requirement for systemic UPR signals, including *bZIP60* transcripts translocation, from treated roots to shoots to elicit the downstream UPR gene *BiP3* in our experimental conditions. We then monitored Tm levels in the shoot and root of seedlings treated with Tm in the shoot–root split culture system, using HPLC/MS. As controls, we

used seedlings exposed to either DMSO (D/D) or Tm (T/T) at both the shoot and the root. As expected, we observed a significant Tm accumulation in the treated shoot and root compared to the respective DMSO (mock control)-only treated tissues (Fig. 3c; D/T and T/D vs D/D), indicating that the seedlings absorbed Tm from the growth medium. Furthermore, the root of seedlings with Tm-treated shoot showed accumulation of Tm (Fig. 3c; T/D vs D/D), indicating that Tm can translocate from the shoot to the root. Importantly however, we found no significant increase of Tm levels in the DMSO only-treated shoot of seedlings with Tm-treated root compared to mock seedlings (Fig. 3c; D/T vs D/D) indicating that Tm is not transported from the root to the shoot in the shoot–root split culture system to a detectable level. Taken together, these results support that an

endogenous UPR signaling acts systemically mainly in a shoot-ward direction that is independent of Tm transport from the root to the shoot.

**Root signals induce UPR genes in unchallenged tissues**. We next performed reciprocal micro-grafting analyses in which the aerial tissue (scion) is grafted onto the root (rootstock) of a different plant, using wild type (Col-0) and *bzip28/60* (Fig. 4). We expected that if an endogenous transcellular UPR signal existed as so far supported by our experiments (Fig. 3, Supplementary Fig. 6), we would observe UPR gene transcripts in the *bzip28/60* tissues grafted with Col-0 tissues. Because our results indicate that a UPR signaling moves mainly in a shoot-ward direction (Fig. 3b, Supplementary Fig. 6), to monitor the UPR signaling at tissue-specific level we compared the abundance of *sbZIP60*, *BiP3* and *bZIP28* transcripts in self-grafts (same genetic background) and hetero-grafts (different genetic background) of scion and rootstock from independent seedlings treated with Tm at the rootstock. As a reference, we used untreated (i.e., DMSO only) micro-grafted seedlings with the same genetic combination as the respective Tm-treated micro-grafted seedlings. As expected, upon Tm treatment the self-grafted *bzip28/60* (scion/rootstock combination indicated as *bzip28/60/bzip28/60*) did not show a significant increase of *sbZIP60*, *BiP3* and *bZIP28* transcripts compared to the same untreated background both in the scion and in the rootstock (Fig. 4a–c). In net contrast, compared to the *bzip28/60/bzip28/60* self-grafts, the Col-0 self-grafts (*Col-0/Col-0*) displayed significant induction of the UPR marker genes, *sbZIP60* and *BiP3*, in both scion and rootstock but non-induced levels of *bZIP28*, consistent with a bZIP28 signaling in ER stress mainly mediated at a protein level[9] (Fig. 4a–c). These controls indicate that the micro-grafting approach does not hamper the response of the grafted unions to ER stress. We then tested the *bzip28/60* scion grafted on Col-0 rootstock (*bzip28/60/Col-0*). Compared to Col-0 self-grafts, in the rootstock of *bzip28/60/Col-0* we found similar levels of UPR gene transcripts (Fig. 4a–c), indicating that the Col-0 rootstock in the *bzip28/60/Col-0* grafts can respond to ER stress as the self-grafted *Col-0/Col-0* rootstock. In the scion of *bzip28/60/Col-0* hetero-grafts, the UPR gene transcript levels were significantly higher compared to the scions of *bzip28/60* self-grafts albeit lower when compared to the *Col-0/Col-0* scion (Fig. 4a–c). Because the *bzip28/60* background is unable to evoke the UPR, these results indicate that the scions of *bzip28/60/Col-0* hetero-grafts contain UPR gene transcripts originated from the Col-0 rootstock. These results are consistent with our observations that a shoot-ward signal originated from a Tm-treated tissue can induce the UPR in an unchallenged systemic tissue (Fig. 3b) and findings that the *bZIP28* mRNA can move intracellularly in unstressed conditions[29].

Next, we tested whether UPR signaling other than the canonical bZIP60 and bZIP28 arms could be involved in the systemic UPR response. We conducted a separated reciprocal grafting with Col-0 as the scion and *bzip28/60* as the rootstock (*Col-0/bzip28/60*). We expected that, if other shoot-ward UPR signaling were in place beside the bZIP28 and bZIP60 arms, then the levels of *BiP3* would be affected in the scion of the *Col-0/bzip28/60* hetero-graft. Similar to *bzip28/60* self-grafts, there was no significant induction of *sbZIP60, BiP3,* and *bZIP28* in the rootstocks of *Col-0/bzip28/60* hetero-grafts upon Tm treatment (Supplementary Fig. 7A–C). Also, in the scions of the *Col-0/ bzip28/60* hetero-grafts the *sbZIP60, BiP3,* and *bZIP28* were not induced and their respective mRNA levels were not significantly different compared to the scions of *bzip28/60* self-grafts (Supplementary Fig. 7A–C). While the lack of *BiP3* induction in the scions is likely due to the absence of a functional UPR

machinery in the rootstocks of *Col-0/bzip28/60* hetero-grafts, these data indicate that the systemic UPR signaling requires the function of the canonical UPR bZIP-arms.

**Systemic UPR signaling is plasmodesmata dependent**. Direct intercellular communication between plant cells, including cells of the stele and the endodermis, occurs via plasmodesmata (PD), which are connecting micro-channels between adjacent cells and the main route for certain signaling molecules in cell-to-cell trafficking[30]. To establish if the systemic UPR relies on PD-mediated traffic, we tested whether sbZIP60 could target the PD. We generated a YFP fusion to sbZIP60 driven by a constitutive promoter for confocal microscopy analyses in leaf epidermal cells, which are more suitable for imaging analyses of PD compared to root tip cells. We verified a nuclear localization of YFP-sbZIP60 as well as a diffused distribution with conspicuous punctate reminiscent of PD (Supplementary Fig. 8A). The PD localization of sbZIP60 was confirmed through co-localization analyses with CFP fused to the established PD-localized receptor-like transmembrane protein, PDLP1[31] (Supplementary Fig. 8A). In contrast, a cytosolic YFP (cYFP) control did not show marked co-localization with PDLP1-CFP (Supplementary Fig. 8A). Integrated density measurements of the YFP signal at the PD using Aniline Blue (AB)[32], a vital PD dye, and Pearson correlation coefficient[33] estimation, as a measure of the association of fluorescence intensity between the YFP and AB signals, indicated that YFP-sbZIP60 co-localized at PD at significantly higher levels compared to cYFP (Supplementary Fig. 8B). The evidence that sbZIP60-GFP moves away from the stele (Fig. 1a, Supplementary Fig. 2), a process that is mediated exclusively by PD, and the coincidental distribution of YFP-sbZIP60 at PD (Supplementary Fig. 8) support that sbZIP60 is translocated via PD in conditions of ectopic expression. A PD dependency of endogenous systemic UPR response was tested in the shoot–root spilt culture system using the conditional PD blockage mutant, *pMDC7-icals3m*[34–36], in which the PD passage is reduced by an enhanced accumulation of callose throughout whole seedling upon gene induction by estrogen[34,37]. In Col-0, in the presence of estrogen, the expression of *sbZIP60*, and *BiP3* was similar to the mock condition, indicating that exogenous estrogen does not elicit ER stress (Fig. 5a). Both UPR markers exhibited similar expression profiles in shoots and roots in conditions of Tm-treatment only, or in condition of Tm-treatment in conjunction with estrogen application to the roots (Fig. 5a), further supporting that the addition of exogenous estrogen does not affect the systemic UPR response. In the conditional PD blockage mutant, there were no significant differences in the transcript levels of UPR markers in Tm+estrogen treated roots compared to Tm only treated roots, indicating no effect of estrogen on the UPR in this mutant (Fig. 5b). Most importantly, the expression of both UPR markers was significantly reduced in the shoots of seedlings treated with Tm+estrogen at the roots compared to the shoots of seedlings with Tm-only treated roots (Fig. 5b). Therefore, an induction of PD closure compromises the systemic UPR signaling likely mediated by a translocation of signaling molecules such as sbZIP60 to elicit UPR in the distal tissues.

## Discussion

Long-distance signaling is required for plants to actuate physiological processes and thrive in response to environmental challenges[38–40]. Here we demonstrate that during a 24 h-time course upon local application of Tm to wild-type seedlings, the UPR markers can be detected in systemic tissues, indicating that ER stress in plants evokes a long-distance signal transduction of the UPR. Our conclusions are further corroborated by reciprocal

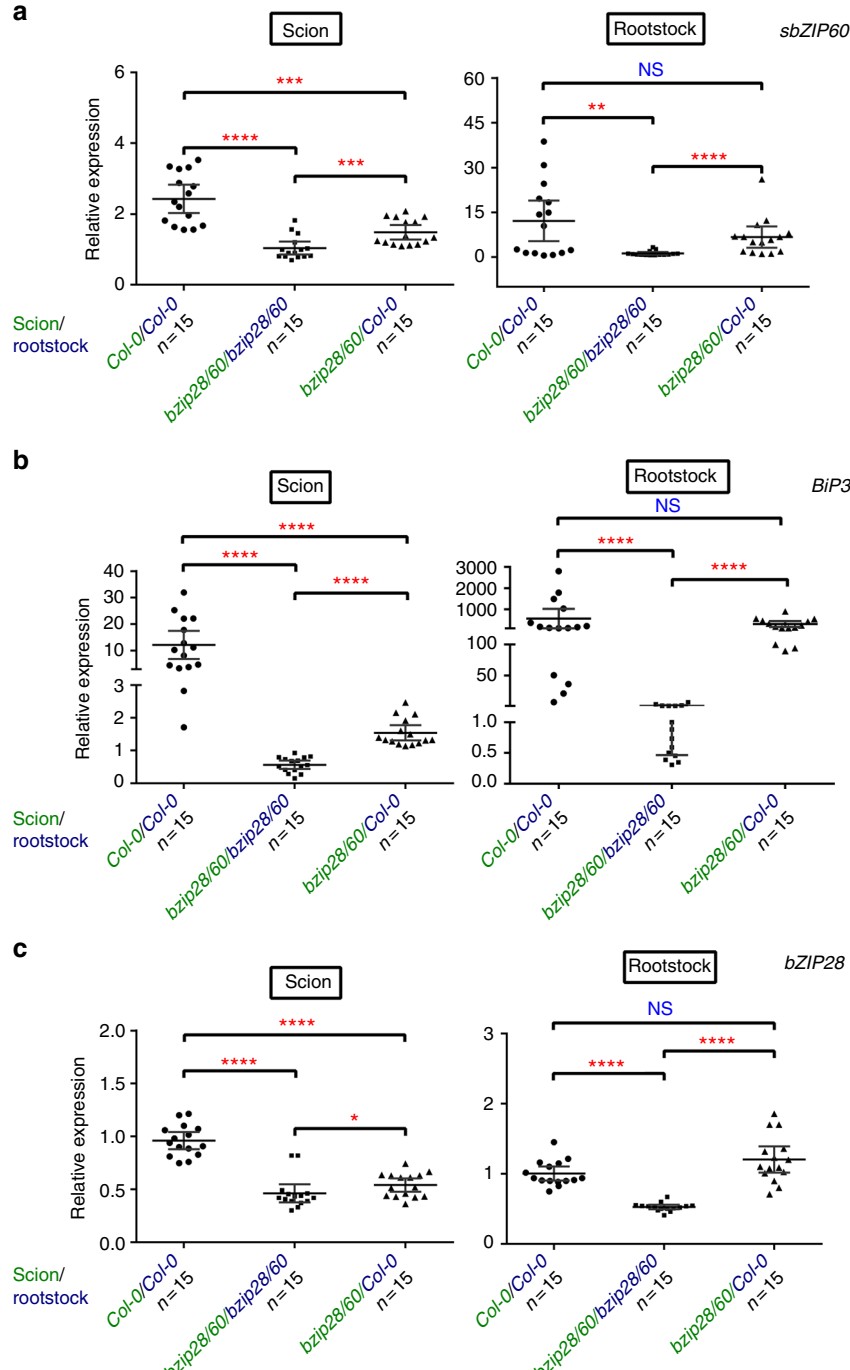

**Fig. 4** Micro-grafting analyses support the existence of endogenous systemic UPR signals. **a–c** Plots of the mean values (dots) of the quantitative RT-PCR analyses of the transcripts of *sbZIP60* (**a**) *BiP3* (**b**) and *bZIP28* (**c**) in scion and rootstock of 4-week-old grafted seedlings of wild-type and *bzip28/60* backgrounds treated with DMSO or 0.5 μM Tm (Tunicamycin) for 48 h as described in Fig. 3a. The average of all the values is indicated by a black line. Error bars represent mean with 95% confidence interval among all grafts. Error bars above and below indicate the 95th and 5th percentiles. Average values significantly different from the values of wild-type (Col-0) and *bzip28/60* self-grafts are indicated by red asterisks (*$P < 0.05$, **$P < 0.01$, ***$P < 0.001$, ****$P < 0.0001$, NS, nonsignificant; Mann-Whitney test). *n*, total number of grafted unions

grafting and ectopic expression assays showing significant transcript levels of *sbZIP60* and *BiP3* distally from the Tm-treated tissues. Therefore, our work demonstrates that, in addition to the well-established existence of a cell-intrinsic UPR signaling, plants harness long-distance signaling to communicate the occurrence of ER stress in a tissue to systemic tissues. We also show that sbZIP60 moves transcellularly and that its movement causes

downstream UPR activation in distal tissues. Based on these results, we propose that sbZIP60 participates as a non-cell autonomous factor to actuate distal UPR signaling directly through its movement across cells.

Previous studies in *C. elegans* and mice relied on overexpression of transcriptionally active XBP1 to infer the existence of systemic UPR signaling[11,12]. Similarly, local expression of

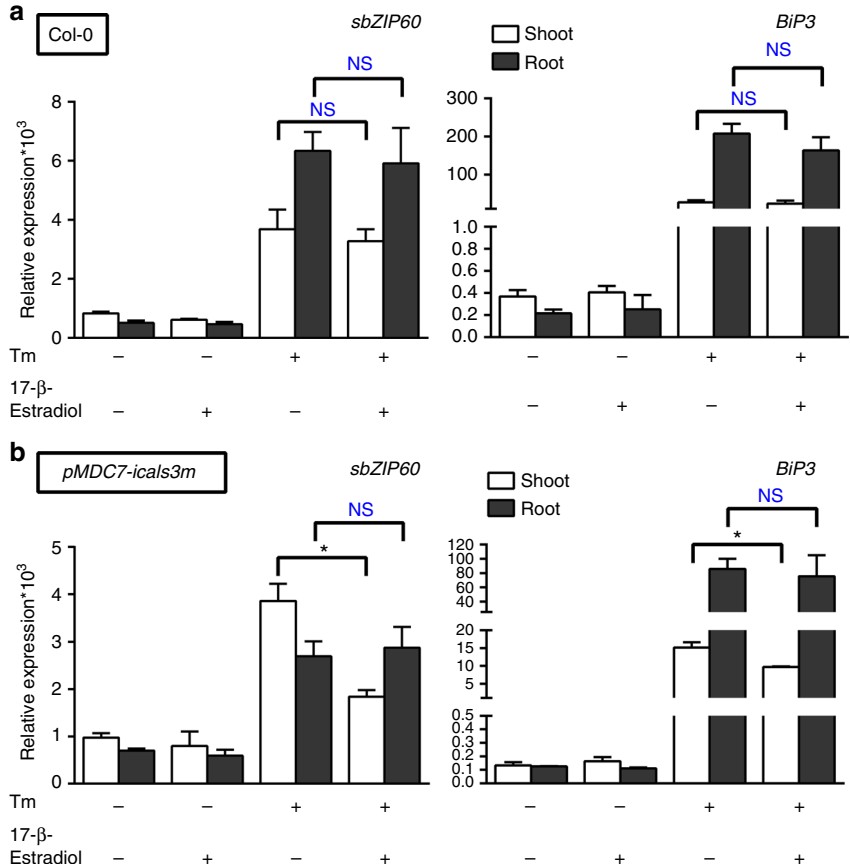

**Fig. 5** Long-distance of UPR signaling relies on the PD availability. **a**, **b** Quantitative RT-PCR analyses of *sbZIP60* and *BiP3* in 14-day-old wild type (Col-0) (**a**) and a conditional PD block mutant (*pMDC7-icals3m*) (**b**) seedlings treated with DMSO (Mock), 10 μM 17-β-estradiol (estrogen), 0.5 μM Tm (Tunicamycin), and 10 μM 17-β-estradiol (estrogen) in combination with 0.5 μM Tm (Tunicamycin) at the root in the shoot–root split system for 24 h. Transcription of *UBQ10* was used as internal control. Error bars represent s.e.m among three biological replicates. Data significantly different from the corresponding control are indicated by asterisks (*$P < 0.05$, NS, nonsignificant; Unpaired *t*-test)

sbZIP60 caused the activation of a downstream target gene. However, an endogenous signal is actuated to evoke a systemic UPR, as supported by the detection of endogenous *sbZIP60* and *BiP3* transcripts in the shoot of wild-type seedlings exposed to Tm at the root in the split-plate system and the micro-grafting experiments using the *bzip28/60/Col-0* hetero-grafts in which we detected the endogenous *sbZIP60* and *BiP3* transcripts (i.e., Col-0-originated) in the scion. The lack of genetic information in the *bzip28/60/Col-0* scions that is necessary to express normally *sbZIP60* and *BiP3* in the aerial tissues supports that the presence of transcripts of these genes in the scion is the result of endogenous systemic UPR signaling. The induction level of *BiP3* in *bzip28/60/Col-0* scion was lower than that in Col-0 shoots under shoot–root split treatment, which may be a consequence of a lack of UPR amplification in *bzip28/60* scions.

In plants, several types of mobile RNA have been identified[41]. Transcriptomic analyses in *Arabidopsis* phloem also reported on the existence of cellular mRNA, suggesting the potential role of these mRNA as signaling molecules in the long-distance trafficking[42]. However, also proteins can contribute to systemic signaling, as it occurs for the well-established floral systemic signaling mediated not only by FT proteins but also by the *FT* mRNA[39]. This may be also the case for sbZIP60 protein and *sbZIP60* mRNA, which both could function as transposable molecules eliciting systemic UPR signaling. The evidence that bZIP60 is localized at the PD supports the possibility that the

bZIP60 protein moves systemically, but it also possible that the intercellular translocation of *bZIP60* mRNA leads to translation of an active transcription factor in systemic tissues.

bZIP28 is another master UPR modulator contributing to the activation of ER stress response genes, including *BiP3*, *ERDJ3A*, and *TIN1* in the plant UPR[19,43]. While the results obtained using *bZIP60* as transgene in the *bzip28/60* background provide evidence for a role of bZIP60 in systemic UPR signaling, the presence of bZIP28 in the wild-type background may facilitate to some extent the expression of *BiP3* in the wild-type shoot of seedlings challenged with Tm at the root on the split-plate system. The *bZIP28* transcripts are transposable between cells, at least in conditions different from ER stress[29]. Therefore, *bZIP28* mRNA or its gene product may be involved in the systemic UPR regulation in parallel to the bZIP60 arm. Indeed, the lack of induction of UPR transcripts in the scions of *Col-0/bzip28/60* hetero-grafts exposed to Tm at the roots argues that the systemic UPR signaling relies on the canonical UPR arms.

In plants, cell-to-cell communication takes place through pathways that involve the apoplast via the continuum of the cell wall, symplastic-driven cytoplasmic transport between different cells within the same tissue or among tissues connected by PD, and vascular-driven transport between different groups of cells or tissues utilizing conducting system composed of phloem and xylem[44]. Using subcellular localization analyses and transcriptional analyses, we have found sbZIP60 protein and *sbZIP60*

mRNA in ectopic tissues, respectively; we also found that the sbZIP60 protein can associate with PD and enter the nucleus, implying that the movement of bZIP60 in systemic signaling may occur through PD. A requirement for PD availability for long-distance UPR signaling is supported by the evidence that in a conditional PD mutant the systemic UPR is attenuated when PD closure is induced. Although we have verified the presence of sbZIP60 protein at the PD, the *sbZIP60* mRNA may translocate through the PD, as it occurs for the mRNA of other proteins[39]. Additionally, the reported existence of unspliced *bZIP60* (*unbZIP60*) mRNA and likely co-localization with ER-associated unbZIP60 protein[20] does not exclude a potential mobility of *unbZIP60* mRNA for involvement in the systemic UPR through PD where a modified ER is present[45]. The visualization of sbZIP60 protein at the PD may be facilitated by expression of sbZIP60 by the CaMV 35S promoter. However, as PD protein targeting is a highly specific process[46], the subcellular localization of sbZIP60 at PD is unlikely a result of overexpression. This is further supported by a relatively lower frequency of localization of cytosolic YFP at the PD in the same experimental conditions. In plants, some systemic response regulators that target PD do not act directly on systemic response effectors[47,48]. The evidence provided in our work that transcellularly translocated sbZIP60 protein can induce the activity of the promoter of a target gene indicates that the systemic movement of sbZIP60 is functional in evoking UPR gene expression.

Biotic stress like pathogen attack induces SAR through which the signals generated from infected sites are translocated to distal plant tissues priming the defense response and eliciting immunity for subsequent infections[40]. Based on the results that sbZIP60 traffics across cells, we propose that, similar to SAR, sbZIP60 participates in long-distance stress signaling to modulate the UPR in cells distally from the site where ER stress occurs. In nature, ER stress is caused by a variety of challenges including pathogens, heat, and salt[7,49,50]. A long-distance signaling of ER stress from challenged tissues may help cells anticipate incoming stress to yet-unchallenged tissues. For example, the UPR is required for plant defense response by modulating secretion of antimicrobial proteins[51]. Therefore, a systemic signaling of ER stress may prepare cells of systemic tissues for responding to a potentially-upcoming ER stress by inducing the accumulation of transcripts of ER stress-attenuating proteins.

In our work, we have verified significant levels of *sbZIP60* and *BiP3* transcript accumulation in the shoot when Tm was applied to the root; conversely, when applied to the shoot, Tm induced *sbZIP60* and *BiP3* transcript accumulation in the roots to much lower levels. Based on these results, we propose that a root-originated transcriptional signal may operate mainly in a shoot-ward direction for ER stress signaling. However, a shoot-originated signal that moves in a root-ward direction may also exist. If such a signal were in place it would operate to lower levels than the root-originated signal, which would prevent its detection by qRT-PCR.

Long-distance stress signaling in plants is mediated by a number of inducers[47]. The role of these inducers in long-distance ER stress signaling is yet to be evaluated and it is conceivable that long-distance ER stress signaling may overlay a bZIP60-mediated signaling with the action of other stress transducers. Nonetheless, it has been shown in seedlings that ER stress responses are independent from endogenous SA and that Tm does not induce accumulation of SA[52]. Therefore, long-distance ER stress signaling is likely independent from SA. The evidence that micro-grafts with a *bzip28/60* rootstock fail to evoke UPR signaling in a Col-0 scion further supports that the systemic UPR signaling mainly relies on the canonical UPR arms.

Our findings address the long-standing question—whether plant UPR constitutes an endogenous systemic signal. From our results, we conclude that this is the case. The identification of bZIP60 as a component of the long-distance UPR signal transduction is a significant step forward in the understanding of the mechanisms underlying systemic signaling transduction of ER stress responses in intact organisms.

## Methods

**Lines and plant growth condition**. *Arabidopsis thaliana* ecotype Columbia-0 (Col-0) was used as a wild-type genotype in this study. The mutant lines used in this work are: *bzip28* (Col-0; SALK_132285), *bzip60-1* (Col-0; SALK_050203)[49]. Sterilized seeds were stratified at 4 °C for 2 days and plated on half-strength Linsmaier Skoog (LS) medium (1/2x LS salts, 1.5% sucrose, 1.2% Agar). Plants were grown in a 16h-light/8h-dark cycle at 21 °C. Agrobacterium (GV3101)-mediated transformation of *Arabidopsis thaliana* and *Nicotiana tabacum* plants was performed by the floral dip and infiltration methods, respectively[53,54].

**Shoot–root split culture system**. Plants were grown vertically on the half-strength LS medium for 14 days and transferred onto a 9 cm Petri dish equipped with two compartments (Kord-Valmark #2903, USA) containing medium with either DMSO in one side or 0.5 μM Tm, 10 μM 17-β-estradiol and 0.5 μM Tm in combination with 10 μM 17-β-estradiol in the other side and cultured horizontally for an additional 1 day. Data were acquired on at least three biological replicates.

**qRT-PCR for gene expression analyses**. Total RNA was extracted using Macherey-Nagel NucleoSpin RNA Plant kit (www.mn-net.com). All samples within an experiment were reverse transcribed at the same time using iScript cDNA synthesis Kit (BIO-RAD 1708891). Real-time qRT-PCR with SYBR green detection was performed in triplicate using the Applied Biosystems 7500 Fast Real-Time PCR System[55]. *UBQ10* was utilized as an internal control in normalization of qRT-PCR, unless otherwise stated in Supplementary Fig. 4. Similar patterns of expression were observed in the three independent biological replicates. Primers used in this study are listed in Supplementary Table 1.

**Tunicamycin measurements**. Two-week-old plate-grown seedlings (∼ 100 mg) exposed to the shoot–root split culture system as detailed above were harvested and ground in liquid nitrogen. The samples were extracted at 4 °C overnight using 1 ml of ice cold methanol:water (80:20 v/v) containing 0.1% formic acid; 0.1 g L$^{-1}$ tunicamycin spiked with propyl 4-hydroxybenzoate as an internal standard for quantification of Tm levels. Then samples were vortexed and centrifuged at 12,000× $g$ 4 °C for 10 min, after which the flow-through samples were transferred to HPLC vials for the measurement of endogenous concentration of Tm by LC-MS with Quattro Premier XE (Waters company) instrument according to an established protocol[56]. The experiments were repeated at least two times with three biological replicates each time showing similar results.

**Grafting experiment**. Hypocotyl micro-reciprocal grafting was performed as described in Marsch-Martínez et al.[57], with minor modifications. Briefly, micro-grafting was conducted using ten-day-old seedlings grown vertically that were first grown in 16h-light/8h-dark 120 μmoles.m$^{-2}$ s$^{-1}$ for 6 days and then in dark for 4 additional days. Grafts were generated by transverse sectioning of the aerial and root portions followed by conjunction on growth medium. After the grafts were made, samples were grown in dark vertically for 2 days, moved to light but covered with fine mesh for 1 day and then uncovered to grow in light. Adventitious roots were removed every 2–4 days. Samples were harvested by removing the whole hypocotyl portion avoiding the potential contamination at grafted junction and processed 4 weeks after micro-grafting for further analysis. Lack of contamination of adventitious roots in grafted plants was confirmed by genotyping with PCR. Analyses were performed on at least three biological replicates.

**Plasmid construction**. The Phusion high-fidelity DNA polymerase (NEB, USA) was employed to amplify all DNA sequences using the primer sets in Supplementary Table 1; the Gateway system (Invitrogen, USA) was used to generate expression plasmids. The promoter sequences of *GLYT*, *SHR*, and *BiP3* and the coding sequences of *spliced bZIP60* and *PDLP1* were amplified using *Arabidopsis* genomic and cDNA as templates, respectively. The amplified coding sequences were recombined into indicated Gateway vectors via LR reaction (Invitrogen, USA) and confirmed by sequencing.

**Confocal laser scanning microscopy imaging**. Confocal imaging was performed with an inverted laser scanning confocal microscope Nikon A1RSi. For subcellular localization assays, leaf tissues were mounted on a slide in a drop of tap water and viewed with the confocal microscope. GFP fluorescence was monitored at excitation wavelength of 488 nm and a bandpass 500–550 nm emission filter. Propidium iodide was monitored with a 560 nm excitation wavelength and a 570–620 nm

bandpass emission filter. YFP fluorescence was monitored with a 513 nm excitation wavelength and 520–550 nm bandpass emission filter. CFP was monitored with a 443 nm excitation wavelength and a 465–505 nm emission filter.

**Quantification of protein co-localization at PD**. *Agrobacterium* cultures at a cell density $A_{600}$: 0.025 and 0.5, containing the constructs for cytosolic YFP and YFP-sbZIP60, respectively, were infiltrated through stomata into *Nicotiana tabacum* leaves using needleless syringe. The images were acquired 48 h after infiltration. Pearson correlation coefficient (PCC) values were calculated using the co-localization tool with default settings of the Nikon microscope software (NIS-Element AR 4.30). Areas with CFP and/or YFP signals at PD puncta were selected for the coefficient calculations. The coefficient calculations were performed on 170 of 3.4 $\mu m^2$ areas.

**Propidium iodide staining**. *Arabidopsis* transgenic seedlings were stained in the 10 $\mu g$ $ml^{-1}$ working solution of propidium iodide which was prepared by diluting with half LS liquid medium from 1 mg $ml^{-1}$ stock solution (Sigma-Aldrich P4864).

**GUS staining**. β-Glucuronidase activity in transgenic roots carrying *pBIP3-GUS* was visualized by X-Gluc as substrate using a conventional protocol[58]. Staining was performed three times in three independent lines with consistent results.

**Aniline Blue staining**. The vital dye Aniline Blue diammonium salt (Sigma-Aldrich 415049) dissolved in 1 M Glycine (pH = 9.5) (0.1% working solution) was adopted for PD-associated callose staining upon stoma infiltration with a needleless syringe into *Nicotiana tabacum* leaves for confocal microscopy analyses. Staining was performed three times in three independent lines with consistent results.

## Data availability
The authors declare that the data supporting the findings of this study are available within the article and its supplementary files. All relevant data are available from the authors upon request.

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

## Acknowledgements

We thank the following colleagues: Dr. P. Benfey for *pSHR-GFP* and *pSHR-SHR-GFP* seeds; Dr. A. Orellana for the seeds of *pbZIP60-GFP-bZIP60*; Dr. K. Gallagher for the seeds of *pMDC7-icalsm*; Dr. L. Chen in the Mass spectrometry and Metabolomics Core Facility at Michigan State University for LC-MS analysis. This work was supported by primarily by the National Institutes of Health (GM101038) with contributing support from the Chemical Sciences, Geosciences and Biosciences Division, Office of Basic Energy Sciences, Office of Science, U.S. Department of Energy (award number DE-FG02-91ER20021), DOE Great Lakes Bioenergy Research Center (DOE BER Office of Science DE-FC02-07ER64494), National Science Foundation (MCB 1714561) and AgBioResearch.

## Author contributions

Y.-S.L. and F.B. designed research; Y.-S.L, G.S., S.Z-D and L.G. performed the research; Y.-S.L., C.R., and F.B. interpreted the results; Y.-S.L and F.B. wrote the manuscript.

## Additional information

**Competing interests:** The authors declare no competing interests.

