## [Peer Review File · Nature Communications]

Reviewers' comments:

Reviewer #1 (Remarks to the Author):

The manuscript titled 'The systemic unfolded protein response in plants is mediated by the transcription factor bZIP60' reports the investigation about whether unfolded protein response (UPR) triggered by ER stress is transferred directly in a long distance via the intercellular movement of bZIP60.

bZIP60, a transcription factor which is normally localized to the ER membrane, has been shown to play a key role in inducing UPR by translocating to nucleus under ER stress. This nuclear localization is enabled by IRE1 via the splicing of bZIP60 mRNA, which removes the coding sequence for a transmembrane domain and exposes nuclear targeting domain instead. Nuclear localized bZIP60 then directly regulates transcription of stress response genes including BiP3. In this manuscript, authors examine the ability of modified version of bZIP60 (sbZIP60), which is targeted to nucleus, in its intercellular trafficking to trigger UPR directly in a long distance.

The key question of this manuscript is whether sbZIP60 acts as a long-distance signal between organs. Authors used four approaches to address this. First, they used SHORTROOT promoter (pSHR) to drive the expression of sbZIP60 fused to GFP (Fig.1) and examined the GFP localization. Second, root specific promoter (pROOT) was used to express sbZIP60 in bzip28 and 60 double mutant (the genetic background that would abolish the UPR) and then the expression of BiP3, a direct target of sbZIP60, was quantified in the shoot and the root (Fig.2). Third, split media assay (tunicamycin treated either shoot or root only) was used to trigger ER stress either only in the root or only in the shoot, and then the expression of BiP3 and sbZIP60 was examined in both shoot and root. Lastly, reciprocal grafting between Col-0 and bzip28/60 mutant was conducted and the BiP3 and sbZIP60 expression was analyzed in the stock and scion.

The overall experimental approaches are sound for determining the mobility of sbZIP60 as intercellular signal. However, I find that some data need further improvement and clarification thus would like to provide comments on these.

Major Comments:

1. Throughout the data supporting the intercellular mobility of sbZIP60, it is not clear whether it is RNA or protein that is mobile between the cell. The plasmodesmal association of sbZIP60 was shown in supplemental figure 1, however the protein does not seem to be targeted efficiently to plasmodesmata. Furthermore, by showing measurement data for sbZIP60 RNA as UPR marker, authors ambiguously state that sbZIP60 is mobile. I would suggest that authors clarify between RNA and protein which is primarily mobile between cells then state the change in UPR markers in a non-cell autonomous manner.

2. Even though sbZIP-GFP proteins show expansion beyond where they are produced as indicated in Fig. 1, how extensive its intercellular mobility is unclear since the confocal microscopy is limited to the meristem. Since the GUS expression under BiP3 promoter is likely via direct upregulation by pSHR - sbZIP60-GFP, sbZIP60-GFP is expected to be found throughout the root, similarly to pBiP3-GUS. Present the figure with confocal imaging of sbZIP-GFP throughout the root.

3. In the measurement of UPR in a long-distance from the initial trigger of ER stress, authors need to explain in the result or materials and methods how tissue sampling for quantitative RT-PCR was performed around the graft junctions.

4. In Figure 2, sbZIP60 level in shoot is higher than Col-0 when it is expressed under root specific promoter. By contrast, Figure 3 shows that sbZIP60 level does not increase noticeably in the shoot when tunicamycin is treated only to the root. I wonder whether the result in the Figure 2 indicates the designated root specific promoter can drive gene expression in the shoot area. Authors showed the root specific expression of GLYT by RT PCR, however this does not necessarily mean that the selected promoter drives gene expression exclusively in the root. In that regard, I think authors need to show

whether pRoot drives gene expression only in the root by using a visual reporter gene. Grafting analysis between Col-0 (scion) and bzip28/60; pRoot::sbZIP60 (stock) as well as grafting between bzip28/60 (scion) and bzip28/60; pRoot::sbZIP60 (stock) will help too.

Minor Comments:

1. In the introduction, it is not explained how the splicing of bZIP60 affects the localization in the cell. This information will help rationalizing the experimental design for sbZIP60.
2. Cell non-autonomous -> non-cell autonomous
3. Translocase -> translocate
4. p.6 Figure S2 should be S3.
5. Correct some errors in the reference formatting
6. Fig. 1: panel A is difficult to interpret because of the lack of cell boundaries. I suggest to use propidium iodide or DIC optics for confocal imaging and overlay with GFP expression. In Panel B, is the expression of pBiP3-GUS in the Col-0 under the ER stress? It would be informative if the authors show the GUS expression in the presence and absence of ER stress.

Reviewer #2 (Remarks to the Author):

In this manuscript the authors assess the non-cell autonomous action of UPR in plants, focusing on bZIP60 with a role in facilitating systemic UPR signaling. However, the proposed hypothesis is not novel, since a role of bZIP60 in stress response has already been reported in previous studies see for example (Iwata and Koizumi., 2005, PNAS). The new findings in this manuscript is the ability of bZIP60 to move to the cortex and epidermis when expressed in the stele in addition to systemic induction of the UPR genes using micrografting experiments.

The idea presented here is interesting and might be relevant in the context of understanding the role of the unfolded protein response (UPR) biotic and abiotic stresses through the intercellular movement bZIP60.

The authors show indeed that bZIP60 moves intercellularly but did not demonstrate the relevance this movement in any context, they reported that the bZIP60 might activate BiP3 in the cells where it moves but this is not enough to illustrate the potential role of bZIP60 as UPR modulator specially that it has already been shown that bZIP60 is involved in plant biotic and abiotic stress responses (Moreno et al., 2012; Plos1), one important question that the authors should have addressed is the importance if bZIP 60 movement in regulating these responses, one way to study this is by blocking bZIP movement and testing plant response to different stresses.

To show non-cell autonomous action bZIP60, the authors expressed the gene under SHR promoter. This experiment is very useful to demonstrate that bZIP is able to move from the stele to the outer tissue layers. However the authors should have established first that the protein moves by comparing its own promoter to the protein, thus the promoter of pbZIP::FP (Fluorescent Protein) is needed to show mRNA accumulation and a complementing pbZIP60:bZIP60::FP (protein localization) in a bzip60 mutant background is critical to assess the endogenous bZIP activity in normal conditions. Then the authors could have subjected these lines to different stresses and monitor their effect on both promoter and protein expression.

In addition the images shown in Figure 1 have poor quality, the cells types are not visible, which makes it difficult to examine the subcellular localization and to recognize different tissue layers, as well as the effect the expressing bZIP60 in the vasculature might have on the root pattern. The authors should use a cell wall dye to improve the visibility of the tissue layers. The roots pictures showing the pSHR-bZIP60-GFP expression are not in the correct focal plane and the images are not taken from the same root region, for example in pSHR-bZIP60-GFP the root cap is missing. Figure 1,

is disorganized and the images showing GUS expression are not presentable even for a pre-submission, the authors should improve the quality of the figure.

The authors show that the bZIP60 colocalize with plasmodesmata, and from there they conclude that "The coincidental distribution of YFP-sbZIP60 and PDL1-CFP supports the possibility that sbZIP60 is translocated between the stele and the endodermis via PD". The authors should have tested this by using lines where plasmodesmata are blocked to show that the protein uses these channels to translocate intercellularly as described in (Vaten et al., 2011, Dev Cell) or using plasmodesmata multiple mutants to show that the protein can move more efficiently in these lines.

The authors are also strongly advised to revise the writing of the manuscript.

Reviewer #3 (Remarks to the Author):

General comments

This is an exciting paper, and this is the first report in plants investigating the long-distance UPR signaling. The primary observation is that bZIP60, a known UPR transducer is involved in cell non-autonomous systemic UPR signaling. To explore the systemic UPR signal transduction, the authors focus on the genetic approach such as using higher order UPR transducers mutant line (bzip28bzip60) and micro-grafting experiments. The authors contend that spliced bZIP60 protein moves transcellularly via PD and regulates downstream UPR signaling. Besides, long-distance ER stress signaling is mediated by sbZIP60 which regulates the expression of BIP3 gene which is a well-known biomarker for the ER stress response. This signaling may be essential for preparing plants for stress response. Authors have performed the experiments in detail and presented the data. Still, needs several key points to be addressed:

1. In Fig.1 and Suppl. Fig.1, authors, claimed that sbZIP60 protein is translocated between the stele and the endodermis via PD where it systemically regulates UPR. In Fig.2-4, authors showed that bZIP60 mRNA is mainly showing long-distance movement from root to shoot and regulate BIP3 gene expression. How authors are sure about only sbZIP60 protein (and not mRNA) movement from cell-to-cell via PD because sbZIP60 mRNA can also translocate to the other cells via PD where it can be expressed after translation?
2. In Fig.1 and Fig.2, authors showed translocation of sbZIP60 protein and mRNA. As it is well known that bZIP28 is also an important transducer of UPR (Liu and Howell, 2007; Iwata and Koizumi, 2012; Howell, 2013). It is reported that the bZIP28 mRNA is cell-to-cell mobile (Thieme et al, 2015, Nat Plants). Thus it needs to check bZIP28 Δ C protein trafficking from cell to cell or long-distance signaling. Please explain that UPR systemic signaling is bZIP60 specific using the bZIP28 as a control.
3. In Fig.3, authors showed that sbZIP60 moves from the root towards shoot under tunicamycin treatment and regulates BIP3 expression. Moreover, shoot treatment with (T/D) accumulates ~10 fold higher levels of tunicamycin in the root tissue than mock, but still, the expression of BIP3 does not induce significantly in the root?
4. In Fig.4, authors used three different graft combinations and gave substantial evidence for root originated sbZIP60 as a signal for systemic UPR signaling. What is expected if Col-0/bzip28/60 grafted plants are treated with Tm? In case, if only sbZIP60 is the only signal responsible for transferring from root to shoot then there will not be any expression of BIP3 gene in the shoot. Another possibility is that the appearance of the BIP3 transcript in shoot indicates that apart from bZIP60 there is something more signaling molecules involved in the process.
5. The splicing of bZIP60 mRNA takes place very fast in response to stress (Deng et al., 2011, PNAS). How about the time required for translocation of sbZIP60 and activation of BIP3 gene expression? Thus, the kinetic response data should be tested and included in the paper.

Minor comments:

1. In discussion, authors need to explain whether it is sbZIP60 protein or mRNA or both are essential for systemic UPR signaling.
2. Comment on whether bZIP60 unspliced mRNA can translocate from root to shoot?
3. Explain "However, BIP3 is not a major target of bZIP2832." However, these papers suggesting otherwise; the bZIP28 binds specifically to ERSE elements and regulate BIP3 expression (Nawkar et al., 2017 PNAS; Songs et al., 2015 PNAS; Zhang et al., 2017 Plant cell;).
4. Page number 6, last line. Fig. S2 should be Fig. S3.
5. Fig. S3. Expression of UBQ10 looks like induced under Tm treatment in the root tissue. It is recommendable to use a qRT-PCR technique with more than two reference genes. Please see Remans et al., 2014 Plant cell.

Reviewer #1 comments:

The manuscript titled ‘The systemic unfolded protein response in plants is mediated by the transcription factor bZIP60’ reports the investigation about whether unfolded protein response (UPR) triggered by ER stress is transferred directly in a long distance via the intercellular movement of bZIP60.

bZIP60, a transcription factor which is normally localized to the ER membrane, has been shown to play a key role in inducing UPR by translocating to nucleus under ER stress. This nuclear localization is enabled by IRE1 via the splicing of bZIP60 mRNA, which removes the coding sequence for a transmembrane domain and exposes nuclear targeting domain instead. Nuclear localized bZIP60 then directly regulates transcription of stress response genes including BiP3. In this manuscript, authors examine the ability of modified version of bZIP60 (sbZIP60), which is targeted to nucleus, in its intercellular trafficking to trigger UPR directly in a long distance.

The key question of this manuscript is whether sbZIP60 acts as a long-distance signal between organs. Authors used four approaches to address this. First, they used SHORTROOT promoter (pSHR) to drive the expression of sbZIP60 fused to GFP (Fig.1) and examined the GFP localization. Second, root specific promoter (pROOT) was used to express sbZIP60 in bzip28 and 60 double mutant (the genetic background that would abolish the UPR) and then the expression of BiP3, a direct target of sbZIP60, was quantified in the shoot and the root (Fig.2). Third, split media assay (tunicamycin treated either shoot or root only) was used to trigger ER stress either only in the root or only in the shoot, and then the expression of BiP3 and sbZIP60 was examined in both shoot and root. Lastly, reciprocal grafting between Col-0 and bzip28/60 mutant was conducted and the BiP3 and sbZIP60 expression was analyzed in the stock and scion. The overall experimental approaches are sound for determining the mobility of sbZIP60 as intercellular signal. However, I find that some data need further improvement and clarification thus would like to provide comments on these.

Major Comments:

1. Throughout the data supporting the intercellular mobility of sbZIP60, it is not clear whether it is RNA or protein that is mobile between the cell. The plasmodesmata association of sbZIP60 was shown in supplemental figure 1, however the protein does not seem to be targeted efficiently to plasmodesmata.

>Authors’ response: We appreciate this comment and performed the experiments to quantify the degree of colocalization of sbZIP60 to PD. The data are shown in Fig. S8B. The results support that sbZIP60 is localized at the PD to a larger extent than cytosolic YFP, which was used as a control. These results support that the protein may be mobile between the cells, although they do not exclude that RNA may also be translocated.

Other cell fate determinants, including FLOWERING LOCUS T (FT), move systemically as protein and mRNA. The results for a translocation of bZIP60 through PD have been strengthened by the inclusion of analyses on a PD-conditional mutant. The analyses show reduced systemic mobility of *bZIP60* mRNA in conditions of reduced PD availability (Fig. 5).

Furthermore, by showing measurement data for sbZIP60 RNA as UPR marker, authors ambiguously state that sbZIP60 is mobile. I would suggest that authors clarify between RNA and protein which is primarily mobile between cells then state the change in UPR markers in a non-cell autonomous manner.

>Authors' response: We clarified it in the text and we think both sbZIP60 protein and *sbZIP60* mRNA are involved in the systemic UPR response. As mentioned above, a translocation of both protein and mRNA has been verified earlier for other systemic factors.

2. Even though sbZIP-GFP proteins show expansion beyond where they are produced as indicated in Fig. 1, how extensive its intercellular mobility is unclear since the confocal microscopy is limited to the meristem. Since the GUS expression under BiP3 promoter is likely via direct upregulation by pSHR-sbZIP60-GFP, sbZIP60-GFP is expected to be found throughout the root, similarly to pBiP3-GUS. Present the figure with confocal imaging of sbZIP-GFP throughout the root.

>Authors' response: We appreciate this comment. We show the pSHR-sbZIP-GFP throughout the root, in the revised Fig. S2.

3. In the measurement of UPR in a long-distance from the initial trigger of ER stress, authors need to explain in the result or materials and methods how tissue sampling for quantitative RT-PCR was performed around the graft junctions.

>Authors' response: Done. We stated it in the materials and methods.

4. In Figure 2, sbZIP60 level in shoot is higher than Col-0 when it is expressed under root specific promoter. By contrast, Figure 3 shows that sbZIP60 level does not increase noticeably in the shoot when tunicamycin is treated only to the root. I wonder whether the result in the Figure 2 indicates the designated root specific promoter can drive gene expression in the shoot area. Authors showed the root specific expression of GLYT by RT PCR, however this does not necessarily mean that the selected promoter drives gene expression exclusively in the root. In that regard, I think authors need to show whether pRoot drives gene expression only in the root by using a visual reporter gene. Grafting analysis between Col-0 (scion) and *bzip28/60*; pRoot::sbZIP60 (stock) as well as

grafting between bzip28/60 (scion) and bzip28/60; pRoot::sbZIP60 (stock) will help too.

>**Authors' response:** We appreciate this comment. The promoter was implemented upon its characterization in Vijaybhaskar *et al.*, 2008. The published data clearly indicated that this promoter controls the expression of a root-specific glycosyltransferase (*GLYT*). These conclusions were reached using promoter-GUS assay. Our qRT-PCR results to detect *GLYT* transcripts in the shoots or in the roots individually, now shown in revised Fig. S4, support that *pRoot* is active in the root and to a negligible extent in the shoot. Furthermore, we show that Tm does not alter the activity of this promoter in either tissue.

Minor Comments:

1. In the introduction, it is not explained how the splicing of bZIP60 affects the localization in the cell. This information will help rationalizing the experimental design for sbZIP60.

>**Authors' response:** We included it in the introduction at p.3.

2. Cell non-autonomous -> non-cell autonomous

>**Authors' response:** Done.

3. Translocase -> translocate

>**Authors' response:** Done.

4. p.6 Figure S2 should be S3.

>**Authors' response:** Done. Actually, we re-ordered the all supplemental figures but we took into account this comment. Thank you.

5. Correct some errors in the reference formatting

>**Authors' response:** Done.

6. Fig. 1: panel A is difficult to interpret because of the lack of cell boundaries. I suggest to use propidium iodide or DIC optics for confocal imaging and overlay with GFP expression.

>**Authors' response:** We appreciate the comment and performed the experiments using propidium iodide. The PI channel merged with the GFP channel shows clearly the cell boundaries as shown in revised Fig. 1.

In Panel B, is the expression of pBiP3-GUS in the Col-0 under the ER stress? It would

be informative if the authors show the GUS expression in the presence and absence of ER stress.

>Authors' response: We appreciate this comment. The *Col-0; pBiP3-GUS* in Fig.1 is under unstressed condition and we performed the *Col-0; pBiP3-GUS* staining under Tm treatment (ER stress) as shown in revised Fig. S3.

Reviewer #2

In this manuscript the authors assess the non-cell autonomous action of UPR in plants, focusing on bZIP60 with a role in facilitating systemic UPR signaling. However, the proposed hypothesis is not novel, since a role of bZIP60 in stress response has already been reported in previous studies see for example (Iwata and Koizumi., 2005, PNAS).

>Authors' response: We are a bit confused here. We believe there may be a slight miscommunication. While a role of bZIP60 in facilitating UPR signaling is established, as we indicate in the introduction, the hypothesis and findings that bZIP60 facilitates systemic UPR signaling are indeed novel, as also indicated in the next of the paragraph of the reviewer's critique and the introductory paragraph of Reviewer 3.

The new findings in this manuscript is the ability of bZIP60 to move to the cortex and epidermis when expressed in the stele in addition to systemic induction of the UPR genes using micrografting experiments.

The idea presented here is interesting and might be relevant in the context of understanding the role of the unfolded protein response (UPR) biotic and abiotic stresses through the intercellular movement bZIP60.

The authors show indeed that bZIP60 moves intercellularly but did not demonstrate the relevance this movement in any context, they reported that the bZIP60 might activate BiP3 in the cells where it moves but this is not enough to illustrate the potential role of bZIP60 as UPR modulator specially that it has already been shown that bZIP60 is involved in plant biotic and abiotic stress responses (Moreno et al., 2012; Plos1), one important question that the authors should have addressed is the importance if bZIP 60 movement in regulating these responses, one way to study this is by blocking bZIP movement and testing plant response to different stresses.

>Authors' response: As correctly indicated by the reviewer, the role of bZIP60 in ER stress responses is to modulate gene expression. In this work, we show that the bZIP60 pool that is systemically translocated in response to ER stress modulates UPR gene expression. In this context, the reviewer's point is addressed. The evidence that the UPR is systemic in plants and that bZIP60 is a component of the systemic signaling warrants novelty of the work. Testing the relevance of these responses to abiotic and biotic

stresses as in Moreno *et al.*, 2012 should be subject of another work. The scope of this work is to test systemic actuation of the UPR in an intact organism, and investigate the systemic movement of bZIP60 and its role in mounting the UPR in systemic tissues. We hope that the reviewer will appreciate the novelty of the findings. The relevance of these findings to other causes of ER stress should be deferred to specific studies. Indeed, the study of Moreno *et al.* 2012 was a follow up paper from the findings of the Howell and Koizumi's labs that demonstrated first that bZIP60 can evoke the expression of *BiP3* in conditions of ER stress. We expect that our work will be followed by others to test the relevance of the findings to abiotic and biotic stresses causing ER stress.

To show non-cell autonomous action bZIP60, the authors expressed the gene under SHR promoter. This experiment is very useful to demonstrate that bZIP is able to move from the stele to the outer tissue layers. However the authors should have established first that the protein moves by comparing its own promoter to the protein, thus the promoter of pbZIP::FP (Fluorescent Protein) is needed to show mRNA accumulation and a complementing pbZIP60:bZIP60::FP (protein localization) in a bzip60 mutant background is critical to assess the endogenous bZIP activity in normal conditions. Then the authors could have subjected these lines to different stresses and monitor their effect on both promoter and protein expression.

>Authors' response: We see the reviewer's point but the suggested experiment would be feasible only if bZIP60 were expressed exclusively in a specific tissue of the root. The ubiquitous distribution of GFP-bZIP60 verified in a complemented line (pbZIP60::bZIP60-GFP) obtained by Parra-Rojas *J et al.*, 2015 argues that the approach is not feasible (now in Fig. S1). Specifically, we analyzed the root in the presence of DMSO or Tm using a very sensitive confocal microscope and found that in both conditions, bZIP60 is present in all tissues of the root. We would like to point out, as also done in the main text, that our transgenic approach to test whether bZIP60 can move across tissues was intended specifically to follow ectopic expression of bZIP60 to follow its translocation. This point, as also supported by the reviewer, is critical to assess movement of bZIP60.

In addition, the images shown in Figure 1 have poor quality, the cells types are not visible, which makes it difficult to examine the subcellular localization and to recognize different tissue layers, as well as the effect the expressing bZIP60 in the vasculature might have on the root pattern. The authors should use a cell wall dye to improve the visibility of the tissue layers. The roots pictures showing the pSHR-bZIP60-GFP expression are not in the correct focal plane and the images are not taken from the same root region, for example in pSHR-bZIP60-GFP the root cap is missing.

>**Authors' response:** We appreciate this comment and re-acquired the confocal image of pSHR-sbZIP-GFP throughout the root as shown in revised Figs.1, S2. We also did PI staining to improve the visibility of the cell layers.

Figure 1, is disorganized and the images showing GUS expression are not presentable even for a pre-submission, the authors should improve the quality of the figure.

>**Authors' response:** We appreciate this comment and reorganized it as shown in revised Fig.1.

The authors show that the bZIP60 colocalize with plasmodesmata, and from there they conclude that “The coincidental distribution of YFP-sbZIP60 and PDLP1-CFP supports the possibility that sbZIP60 is translocated between the stele and the endodermis via PD”. The authors should have tested this by using lines where plasmodesmata are blocked to show that the protein uses these channels to translocate intercellularly as described in (Vaten et al., 2011, Dev Cell) or using plasmodesmata multiple mutants to show that the protein can move more efficiently in these lines.

>**Authors' response:** It is a good point. We performed the experiment to determine systemic UPR response using the conditional PD block mutant and data is shown in Fig.5. The results are consistent that bZIP60 moves through PDs to evoke the UPR in distal tissues. Thank you for this suggestion.

Reviewer #3 (Remarks to the Author):

General comments

This is an exciting paper, and this is the first report in plants investigating the long-distance UPR signaling. The primary observation is that bZIP60, a known UPR transducer is involved in cell non-autonomous systemic UPR signaling. To explore the systemic UPR signal transduction, the authors focus on the genetic approach such as using higher order UPR transducers mutant line (bzip28bzip60) and micro-grafting experiments. The authors contend that spliced bZIP60 protein moves transcellularly via PD and regulates downstream UPR signaling. Besides, long-distance ER stress signaling is mediated by sbZIP60 which regulates the expression of BIP3 gene which is a well-known biomarker for the ER stress response. This signaling may be essential for preparing plants for stress response. Authors have performed the experiments in detail and presented the data. Still, needs several key points to be addressed:

1. In Fig.1 and Suppl. Fig.1, authors, claimed that sbZIP60 protein is translocated between the stele and the endodermis via PD where it systemically regulates UPR. In Fig.2-4, authors showed that bZIP60 mRNA is mainly showing long-distance

movement from root to shoot and regulate BiP3 gene expression. How authors are sure about only sbZIP60 protein (and not mRNA) movement from cell-to-cell via PD because sbZIP60 mRNA can also translocate to the other cells via PD where it can be expressed after translation?

>Authors' response: Sorry for the ambiguity. We clarified in the text that the data do not distinguish whether either the bZIP60 protein or mRNA is able to move between cells. Other cell fate determinants, including FLOWERING LOCUS T (FT), move systemically as protein and mRNA. We can detect *bZIP60* transcripts in systemic tissues but we can also detect bZIP60 protein at the plasmodesmata. Distinguishing between the two possibilities is not critical because either scenario or both would support systemic translocation of bZIP60. We have amended the text to clarify this point.

2. In Fig.1 and Fig.2, authors showed translocation of sbZIP60 protein and mRNA. As it is well known that bZIP28 is also an important transducer of UPR (Liu and Howell, 2007; Iwata and Koizumi, 2012; Howell, 2013). It is reported that the bZIP28 mRNA is cell-to-cell mobile (Thieme et al, 2015, Nat Plants). Thus it needs to check bZIP28 Δ C protein trafficking from cell to cell or long-distance signaling. Please explain that UPR systemic signaling is bZIP60 specific using the bZIP28 as a control.

>Authors' response: We agree that *bZIP28* mRNA has been found in hetero-grafts between two Arabidopsis accessions, although the Thieme *et al.* work was not performed to test systemic UPR. In this study, we focused and established the role of bZIP60 in the systemic UPR signaling as a tool to probe for the existence of a systemic UPR in plants. We do not exclude the existence of other signals in such systemic UPR response. However, following the suggestions, we performed the proposed micro-grafting experiments and we detected a trace of *bZIP28* transcripts in the unstressed scion of *bzip28/60/Col-0* hetero-grafts as shown in Fig. 4C. The results are consistent with the work of Thieme *et al.* that *bZIP28* transcripts are mobile. Therefore, *bZIP28* may also be involved in the systemic regulation. We have modified the discussion accordingly and kept the focus of the work on the concept that a systemic UPR exists in plants and that bZIP60 is a signaling component of the systemic signaling.

3. In Fig.3, authors showed that sbZIP60 moves from the root towards shoot under tunicamycin treatment and regulates BIP3 expression. Moreover, shoot treatment with (T/D) accumulates ~10 fold higher levels of tunicamycin in the root tissue than mock, but still, the expression of BIP3 does not induce significantly in the root?

>Authors' response: The induction of *BiP3* is not proportional to the content of Tm inside the plants. However, it is significant with $P < 0.05$.

4. In Fig.4, authors used three different graft combinations and gave substantial evidence for root-originated sbZIP60 as a signal for systemic UPR signaling. What is expected if Col-0/bzip28/60 grafted plants are treated with Tm? In case, if only sbZIP60 is the only signal responsible for transferring from root to shoot then there will not be any expression of BIP3 gene in the shoot. Another possibility is that the appearance of the BIP3 transcript in shoot indicates that apart from bZIP60 there is something more signaling molecules involved in the process.

>Authors' response: This is a very good point. We performed the *Col-0/bzip28/60* grafting combination and we did not detect the presence of *BiP3* transcript in the unstressed scion. The new data are shown in Fig. S7. The results support that the canonical UPR arms are required for the systemic UPR signaling, but unlike the reviewer's suggestion, cannot exclude that bZIP28 also contributes to the signaling. Nonetheless, the comment was very helpful and allowed us to establish that the systemic UPR depends on the canonical UPR signaling arms. We have included these points in the revised discussion. Thank you.

5. The splicing of bZIP60 mRNA takes place very fast in response to stress (Deng et al., 2011, PNAS). How about the time required for translocation of sbZIP60 and activation of BIP3 gene expression? Thus, the kinetic response data should be tested and included in the paper.

>Authors' response: It is a great point. We conducted the experiment to establish the kinetic profile of *sbZIP60* and *BiP3* transcripts in the systemic ER stress response in the shoot-root split culture system. We detected the presence of *sbZIP60* first at 3 hrs post ER stress followed by induction of *BiP3* at 6 hrs post ER stress in the unstressed shoots. Data are shown in Fig. S6.

Minor comments:

1. In discussion, authors need to explain whether it is sbZIP60 protein or mRNA or both are essential for systemic UPR signaling.

>Authors' response: Done.

2. Comment on whether bZIP60 unspliced mRNA can translocate from root to shoot?

>Authors' response: In Parra-Rojas J *et al.*, 2015, the authors detected the mRNA corresponding to the unspliced form of *bZIP60* using capillary electrophoresis coupled to laser induced fluorescence (CE-LIF) indicating the existence of unspliced *bZIP60* mRNA. Meanwhile, they also detected the ER localization of unspliced bZIP60-GFP protein. Due to the co-translation machinery of ER membrane protein synthesis, the unspliced *bZIP60* mRNA may attach to ER therefore it is possible unspliced *bZIP60*

could pass through PD passage, as a modified ER is inside PD. These comments have been included in the revised discussion.

3. Explain “However, BIP3 is not a major target of bZIP2832.” However, these papers suggesting otherwise; the bZIP28 binds specifically to ERSE elements and regulate BIP3 expression (Nawkar et al., 2017 PNAS; Songs et al., 2015 PNAS; Zhang et al., 2017 Plant cell;).

>**Authors’ response:** We are sorry for the misinterpretation. Both bZIP60 and bZIP28 binds to the ERSE element of *BiP3* promoter contributing to *BiP3* expression. However, it has been recently shown that *BiP3* expression is dependent more on bZIP60 than bZIP28; on the other hand, other genes are more dependent on bZIP28 compared to bZIP60 (Ruberti *et al.*, 2018, The Plant Journal).

4. Page number 6, last line. Fig. S2 should be Fig. S3.

>**Authors’ response:** Done. Actually, we re-ordered the all supplemental figures.

5. Fig. S3. Expression of UBQ10 looks like induced under Tm treatment in the root tissue. It is recommendable to use a qRT-PCR technique with more than two reference genes. Please see Remans et al., 2014 Plant cell.

>**Authors’ response:** Thank you. We replaced the original RT-PCR analyses with qRT-PCR using three reference genes (i.e., *ACT8*, *UBQ10* and *IPP2*). We confirmed the original observations. The data are shown in revised Fig. S4.

Reviewers' comments:

Reviewer #1 (Remarks to the Author):

Authors of the revised manuscript titled 'A systemic signaling contributes to the unfolded protein response of the plant endoplasmic reticulum' addressed the most of comments made in the previous submission. Authors tried to show that sbZIP60 serves as mobile factor to trigger the UPR in long distance by demonstrating the presence of sbZIP60 mRNA and sbZIP60 proteins as well as its target BiP3 beyond their original expression domains. However, as indicated in Introduction of this manuscript, bZIP60 is naturally transcribed throughout the plant and the alternative splicing which directs the resulting protein to the nucleus triggers the UPR. In addition, its transcription is induced under ER stress, likely resulting in amplification of bZIP60 via positive feedback loop. In that context, it is important for authors to show how significant the systemic signaling mediated via sbZIP60 is in vivo. In this revision, I find following additional questions about the data from artificially setup experiments and authors' interpretation.

1. Figure S1: bZIP60 protein seems to be targeted to the nucleus even in the absence of ER stress though its level is lower than the one under the ER stress. How does this reconcile with the expression patterns of BiP3?

2. Page 4: We used the SHR promoter to drive expression of cytosolic green fluorescent protein (GFP) (pSHR-GFP)²⁰,

Comment: Do you mean endoplasmic reticulum targeted GFP here?

3. Page 8: In the untreated shoots, a significant splicing of bZIP60 was detected at 3 hrs post treatment and BiP3 expression was predominantly detected at 6 hrs post treatment (Figure S6A,B) supporting a roughly 3 hrs-requirement for systemic UPR signals, including sbZIP60 transcripts translocation, from treated roots to shoots to elicit the downstream UPR gene BiP3 in our experimental conditions.

Comment: This supplementary figure 6 does not specifically compare the spliced version of bZIP60 with the unspliced version. This sentence indicates the splicing as the primary response that results in the increase of the sbZIP60 transcript. However, the data do not distinguish whether the increase of sbZIP60 is from the splicing or from the induction of transcription.

4. Figure 5: In the paper about pMDC7-icals3m (Wu et al. 2016), they designed callose accumulation in a cell type specific manner. It is not clear to me which promoter is driving icals3m in this study.

5. Unlike shoot-root split culture on Col-0 in Fig. 3, the magnitude of change of BiP3 in the shoot (scion) of bzip60/28 grafted on Col-0 seems quite small (Fig. 4). In my opinion, this indicates that systemic response directed by the mobile sbZIP60 might play minor role in comparison to the signal propagation mediated by transcriptional and posttranscriptional upregulation of sbZIP60 in response to ER stress

Reviewer #3 (Remarks to the Author):

I am pleased to inform you that the authors gave answers for all the comments that I have raised. Thus, quality of the paper is good enough to be published in this journal.

We would like to thank both reviewers for their support as well as time for considering our manuscript. As detailed below, we amended all the points raised by reviewer 1, which helped clarify aspects of the work.

Reviewers' comments:

Reviewer #1 (Remarks to the Author):

Authors of the revised manuscript titled ‘A systemic signaling contributes to the unfolded protein response of the plant endoplasmic reticulum’ addressed the most of comments made in the previous submission. Authors tried to show that sbZIP60 serves as mobile factor to trigger the UPR in long distance by demonstrating the presence of sbZIP60 mRNA and sbZIP60 proteins as well as its target BiP3 beyond their original expression domains. However, as indicated in Introduction of this manuscript, bZIP60 is naturally transcribed throughout the plant and the alternative splicing which directs the resulting protein to the nucleus triggers the UPR. In addition, its transcription is induced under ER stress, likely resulting in amplification of bZIP60 via positive feedback loop. In that context, it is important for authors to show how significant the systemic signaling mediated via sbZIP60 is in vivo. In this revision, I find following additional questions about the data from artificially setup experiments and authors’ interpretation.

1. Figure S1: bZIP60 protein seems to be targeted to the nucleus even in the absence of ER stress though its level is lower than the one under the ER stress. How does this reconcile with the expression patterns of BiP3?

>Authors’ response: Indeed, under non-stress conditions, we detected a low level expression of *BiP3* in the roots of 11-day-old seedlings by GUS assay (Fig. 1C) consistent that a basal level of UPR is enacted under physiological conditions of growth to facilitate protein folding.

2. Page 4: We used the SHR promoter to drive expression of cytosolic green fluorescent protein (GFP) (pSHR-GFP)²⁰,

Comment: Do you mean endoplasmic reticulum targeted GFP here?

>Authors’ response: The GFP mentioned here is not endoplasmic reticulum targeted. The GFP coding region is fused to the SHR promoter as result when SHR promoter is activated the GFP protein localizes in the nucleoplasm and cytoplasm as stated in the Haseloff et al., 1997 that is consistent with cytosolic pattern we observed in Fig1A. We also included the reference (ref 16).

3. Page 8: In the untreated shoots, a significant splicing of bZIP60 was detected at 3

hrs post treatment and BiP3 expression was predominantly detected at 6 hrs post treatment (Figure S6A,B) supporting a roughly 3 hrs-requirement for systemic UPR signals, including sbZIP60 transcripts translocation, from treated roots to shoots to elicit the downstream UPR gene BiP3 in our experimental conditions.

Comment: This supplementary figure 6 does not specifically compare the spliced version of bZIP60 with the unspliced version. This sentence indicates the splicing as the primary response that results in the increase of the sbZIP60 transcript. However, the data do not distinguish whether the increase of sbZIP60 is from the splicing or from the induction of transcription.

>Authors' response: We appreciate this comment. To address the point, we examined the kinetics of the expression of the unspliced form of *bZIP60*. Consistent with an increase of sbZIP60 form, we found a significant induction of *unbZIP60* in non-stressed shoots at 3 hrs post Tm treatment. Data are shown in revised supplementary Fig. 6A. The unbZIP60 is likely originated from either Tm-stressed roots or induced by transposable sbZIP60 reaching the shoots. It is also possible that the increased sbZIP60 is due to both enhanced splicing and transcription as a result from a self-amplification loop. Independently from these possibilities, the increase in spliced bZIP60 transcripts is consistent with the observed kinetics of *BiP3* transcription and spliced *bZIP60* transcript increase.

4. Figure 5: In the paper about pMDC7-icals3m (Wu et al. 2016), they designed callose accumulation in a cell type specific manner. It is not clear to me which promoter is driving icals3m in this study.

>Authors' response: We apologize for the lack of explanation. Unlike the cell specific promoter (*EN7*) (Wu et al. 2016) used for endodermis-specific accumulation of callose, we used pMDC7 promoter which is an estrogen inducible promoter expressed throughout the seedlings as described in Curtis, M. D et al., 2003 and Zuo et al., 2000 (ref 35,37). These references were included in the original submission and we have amended the text to provide a clarification. Thank you.

5. Unlike shoot-root split culture on Col-0 in Fig. 3, the magnitude of change of BiP3 in the shoot (scion) of *bzip60/28* grafted on Col-0 seems quite small (Fig. 4). In my opinion, this indicates that systemic response directed by the mobile sbZIP60 might play minor role in comparison to the signal propagation mediated by transcriptional and posttranscriptional upregulation of sbZIP60 in response to ER stress.

>Authors' response: We appreciate this comment and we completely agree. As detailed in the discussion, the systemic UPR may prime distal tissues to prepare for an upcoming stress. We also do not exclude the possibility that other signals may be involved in the systemic UPR activation. Another possible explanation is

that the small but significant induction of *BiP3* in the scion of *bzip60/28* grafted on Col-0 may due to the lack of UPR amplification machinery in *bzip60/28* scion. We also included these explanations in the discussion.

Reviewer #3 (Remarks to the Author):

I am pleased to inform you that the authors gave answers for all the comments that I have raised. Thus, quality of the paper is good enough to be published in this journal.

>Authors' response: Thank you.